# Is There Evidence for IGF1R-Stimulating Abs in Graves’ Orbitopathy Pathogenesis?

**DOI:** 10.3390/ijms21186561

**Published:** 2020-09-08

**Authors:** Christine C. Krieger, Susanne Neumann, Marvin C. Gershengorn

**Affiliations:** Laboratory of Endocrinology and Receptor Biology, National Institute of Diabetes and Digestive and Kidney Diseases, National Institutes of Health Bethesda, Bethesda, MD 20892, USA; christine.krieger@nih.gov (C.C.K.); SusanneN@intra.niddk.nih.gov (S.N.)

**Keywords:** IGF1R, TSHR, receptor crosstalk, Graves’ orbitopathy, IGF1R antibodies, TSHR antibodies, Graves’ orbital fibroblasts, hyaluronan

## Abstract

In this review, we summarize the evidence against direct stimulation of insulin-like growth factor 1 receptors (IGF1Rs) by autoantibodies in Graves’ orbitopathy (GO) pathogenesis. We describe a model of thyroid-stimulating hormone (TSH) receptor (TSHR)/IGF1R crosstalk and present evidence that observations indicating IGF1R’s role in GO could be explained by this mechanism. We evaluate the evidence for and against IGF1R as a direct target of stimulating IGF1R antibodies (IGF1RAbs) and conclude that GO pathogenesis does not involve directly stimulating IGF1RAbs. We further conclude that the preponderance of evidence supports TSHR as the direct and only target of stimulating autoantibodies in GO and maintain that the TSHR should remain a major target for further development of a medical therapy for GO in concert with drugs that target TSHR/IGF1R crosstalk.

## 1. Introduction

Graves’ orbitopathy (GO) (also termed Graves’ ophthalmopathy, thyroid-associated ophthalmopathy or thyroid eye disease) has been under intense study with a goal to not only understand its pathogenesis but also to aid in the development of medical therapies. GO pathophysiology appears to involve activation of receptors for thyrotropin (thyroid-stimulating hormone, TSH) and insulin-like growth factor 1 (IGF1) on fibroblasts/preadipocytes in orbital tissue (OFs) [1]. Under normal physiological conditions, TSH receptors (TSHRs), which are G protein-coupled receptors (GPCRs or seven transmembrane-spanning receptors) and IGF1 receptors (IGF1Rs), which are receptor tyrosine kinases (RTKs), are cell-surface proteins that are activated by their respective ligands that, in turn, are themselves under tight regulation. Persistent activation of GPCRs and RTKs is involved in several diseases. For example, activating mutations of TSHRs and IGF1Rs have been shown to cause hyperthyroidism [2] and tumor formation [3], respectively. In autoimmune diseases, such as Graves’ disease (GD), autoantibodies have been found in the circulation that activate receptors consistently. Lack of physiologic regulation leads to overstimulation and disease [4,5]. In GO, a consensus has emerged that there are stimulating autoantibodies that bind to and directly activate TSHRs (TSAbs) on OFs [4,6,7]. However, it remains controversial as to whether there are autoantibodies that bind to and directly activate IGF1Rs (stimulating IGF1RAbs) [8,9]. It appears that cell-mediated immunity plays a role in GO, and it is possible that there are TSHRs and IGF1Rs on lymphocytes that may be involved; however, these receptors would have to be activated by the antibodies.

As noted above, a number of studies have found that both TSHRs and IGF1Rs are involved in GO pathogenesis [1,10]. These studies have been performed primarily in vitro using OFs from GO patients (GOFs) or fibroblast-like cells isolated from the blood of GO patients (so-called fibrocytes) [9]. Of note, the majority of these studies did not attempt to show direct activation of IGF1Rs but instead relied on the ability of immunoglobulins from GO patients (GO-Igs) to either replicate some of the post-receptor effects of IGF1 or have their effects inhibited by IGF1R antagonists. In only three previous reports were effects of GO-Igs on the IGF1R studied [11,12,13] (see below). An alternative possibility for the involvement of IGF1Rs in GO pathogenesis is that IGF1R involvement results not from stimulating IGF1RAbs but from crosstalk of IGF1Rs with TSHRs that are activated by TSAbs [11].

In this review, we first contrast what is known about TSHR autoantibodies in GD hyperthyroidism and GO to what has been found regarding IGF1R autoantibodies. We describe a model of TSHR/IGF1R crosstalk and present evidence that IGF1R’s role in GO could be explained by this mechanism. Then, we evaluate the evidence for and against IGF1R as a direct target of stimulating IGF1RAbs in GO pathogenesis. Our conclusion that GO pathogenesis does not involve directly stimulating IGF1RAbs will be based, in part, on negative data. Although a negative hypothesis cannot be conclusively proved, we reason that the preponderance of evidence supports TSHR as the direct and only target of stimulating autoantibodies in GO.

## 2. The Role of TSHR and IGF1R Autoantibodies in GO Pathogenesis

Although autoantibodies are a major contributing factor to the development of GO, the mechanism through which this occurs is unclear. One hypothesis proposes that shedding of the A-subunit of TSHR leads to the initial autoimmune response including generation of TSAbs. In this scenario, recently reviewed by Rapoport et al. [14], TSHR A-subunits are cleaved from the extra-cellular domain and released into the lymphatic system. Because of its highly glycosylated state, the A-subunit is taken up by antigen-presenting cells. Resulting B-cell proliferation and maturation may lead to the direct cause of GD and GO. This hypothesis is supported by recent efforts to produce animal models for GD and GO [15,16] (reviewed in [17]), which used the human TSHR A-subunit to generate TSAbs in mice. McLachlan et al. further demonstrated that mouse TSHR A-subunit, which was less glycosylated than human, was unable to induce production of TSAbs. These and similar works were instrumental in our current understanding of GD and GO.

To our knowledge, no group has attempted to create an animal model of GO using IGF1R as an immunogenic agent. It stands to reason that if GO were a disease induced by IGF1RAbs, this strategy could be employed to create an experimental system of significant value. Such an animal model could aid in understanding GO pathogenesis. It would also facilitate testing of future therapies targeting IGF1R, if IGF1RAbs were the source of orbitopathy. However, as helpful as this animal model may be, the work in human patients investigating the putative role of IGF1RAbs in GO pathogenesis is still in beginning stages.

Studying autoantibodies in patients is far from straightforward. Even after isolating antibodies for their specific antigen, they are polyclonal, and their function ranges from stimulating, to neutral, to antagonizing. Antibody populations also change during the progression of disease, as has been documented for TSHR antibodies in GD and GO patients [18,19]. Nevertheless, development of assays that assess function as well as measure titer of TSHR antibodies has greatly aided our ability to use TSHR antibodies as biomarkers of GD and GO (reviewed in [20]).

In contrast, investigation and characterization of IGF1RAbs from GO patients have not seen similar progress. Evidence for IGF1RAbs in GO patients was first reported by Weightman et al. in 1993 [21]. In 2013, two contemporary studies set the stage for the still unresolved controversy as to the prevalence and function of IGF1RAbs in GO patients. Minich et al. reported the presence of IGF1RAbs in a subset of GO patients and further characterized the biological function of those IGF1RAbs, finding them to be antagonistic [12]. In contrast, Varewijk et al. found that GO patients with high TSAb titers were more likely to have stimulatory IGF1RAbs [13]. These reports conflicted on the function of IGF1RAbs but employed different methods, precluding direct comparisons.

More recently, Marino et al. [22] and Lanzolla et al. [23], both with the University Hospital of Pisa, have measured serum IGF1RAbs in GD and GO patients using a newly available commercial assay for IGF1R antibodies. Unlike the assay used by Varewijk et al., the sandwich ELISA used by Marino and colleagues had the advantage of directly measuring circulating IGF1RAbs and distinguishing them from other antibodies [22]. First, Marino et al. compared the incidence of IGF1RAbs in patients with GD (hyperthyroid), patients with autoimmune thyroiditis (hypothyroid), and healthy subjects (no past or present thyroid disease). Within the GD group, 54/80 subjects had GO. Patients with GD were more likely than other groups to have IGF1RAbs positive sera, however, the presence of IGF1RAbs was not significantly different in patients with GO compared to GD patients without GO. Within the patients with IGF1RAb positive sera, Marino et al. next compared IGF1RAb concentrations amongst groups. Again, GD patients had higher concentrations, but no significant difference was found between GD patients with or without GO. Surprisingly, IGF1RAb concentration negatively correlated with GO severity. The authors inferred those IGF1RAbs were antagonistic, though this was not tested in a functional assay.

Lanzolla et al. more deeply investigated possible differences between GD patients with or without GO by measuring serum IGF1RAbs in the same patients before and after radioiodine treatment [23]. Here, IGF1RAb concentrations were higher in GD patients compared to GO. Furthermore, IGF1RAb concentration did not correlate with GO incidence or severity. The authors noted that concentration of IGF1RAbs negatively correlated with TSAb titers in GO patients, though the clinical significance of this finding is open to further study.

In the above reports [12,13,22,23], sera from healthy subjects were positive for IGF1RAbs, and data from these healthy subjects were used to establish a normal reference value. Under ideal circumstances, this value would be determined using a larger population in order to accurately assess the prevalence of IGF1RAbs in GO and other diseases. A confounding factor in the study by Lanzolla was that only one patient had an IGF1RAb concentration greater than the cutoff previously established by Marino et al. According to the manufacturer of the ELISA, none of the GO and almost none of the GD patients had IGF1RAb positive serum. Given the small number of healthy subjects Marino et al. used to establish the normal reference value (*n* = 27), it is entirely possible that those individuals happened to have abnormally high IGF1RAb levels. We think additional studies are needed to confirm this point.

GD and GO are likely initiated by an immune response against TSHR. If IGF1RAbs are produced during later stages of GO, they have not been shown to be pathogenic. In fact, no stimulatory IGF1RAb has ever been demonstrated. Yet these conclusions are not definitive because of lack of data, highlighting the urgent need for continued research.

## 3. Evidence of TSHR/IGF1R Crosstalk

A functional relationship between TSHR and IGF1R signaling was initially established in thyroid cells in which simultaneous activation of the two receptors by their respective ligands was shown to cause synergistic increases in DNA synthesis and cell proliferation [24,25,26]. We [27] made a similar observation in GOFs by showing simultaneous activation by TSH and IGF1 caused synergistic stimulation of hyaluronan (HA, hyaluronic acid) secretion. Increased HA secretion by GOFs is an important part of the pathogenesis of GO.

Other evidence supporting a role for interactions between TSHRs and IGF1Rs was that IGF1R antagonists inhibited TSH signaling. Tsui et al. [28] showed that an IGF1R blocking antibody, 1H7, inhibited TSH activation of mitogen-activated protein kinase 1 (MAPK or ERK) in thyrocytes. We [27] showed that the small molecule IGF1R inhibitor linsitinib inhibited TSH stimulation of HA secretion. We found that this interaction was bidirectional because a TSHR antagonist, ANTAG3 (NCGC00242364), was able to partially inhibit IGF1-stimulated HA secretion, similar to bidirectional GPCR/RTK crosstalk observed in other systems [29].

The finding that 1H7 inhibits TSH activation of MAPK in thyrocytes [28] is consistent with the idea that the signaling interaction between TSHR and IGF1R occurs in the upstream part of the signaling pathway. To show directly that activation of MAPK was caused by TSHR/IGF1R crosstalk, We [8] determined that TSH and IGF1 synergized to activate MAPK in GOFs and that an inhibitor of the MAPK cascade inhibited HA secretion by GOFs. Moreover, in HEK293 cells stably overexpressing TSHRs, we showed that MAPK activation by TSH was dependent on Gi/o proteins [30]. These findings demonstrate that TSHR/IGF1R crosstalk occurs proximal to the receptors rather than downstream in the signal transduction pathway.

These observations raised the question of whether TSHR/IGF1R crosstalk was dependent on a close physical association between these receptors. The earlier findings that TSHR and IGF1R co-localized by fluorescence microscopy and were co-immunoprecipitated [28] were consistent with the idea that these receptors were physically associated, but did not demonstrate that they were coupled functionally. We found that simultaneous stimulation of TSHR and IGF1R leads to synergistic activation of MAPK. This suggested that β-arrestin may mediate TSHR/IGF1R crosstalk based on reports by Boutin et al. [31], showing that MAPK activation by TSH in U2OS cells stably overexpressing TSHRs was dependent on β-arrestin 1. It has been shown that β-arrestin 1 was recruited to IGF1R upon IGF1 binding [32] and that it is involved in several downstream signaling pathways including ERK pathways [33]. Moreover, GPCR/RTK crosstalk in other cell systems involves β-arrestins as well [34]. A potential role of β-arrestin 1 and β-arrestin 2 in TSHR/IGF1R crosstalk was tested by siRNA knockdown in GOFs [35]. β-arrestin 1 knockdown markedly inhibited HA secretion stimulated by the human monoclonal TSHR-stimulating antibody M22, the mouse monoclonal TSHR-stimulating antibody KSAb1 [36] and GO-Igs, and by TSH plus IGF1 by 20–60%. Using a proximity ligation assay (PLA), we showed that TSHR and IGF1R were within 40 nanometers of each other and that this proximity was disrupted when β-arrestin 1 expression was reduced. β-arrestin 2 knockdown did not show this effect. Thus, we conclude that TSHR/IGF1R crosstalk occurs in GOFs and is dependent on the close proximity of these receptors scaffolded by β-arrestin 1.

## 4. Evidence that GO-Igs Activate TSHR/IGF1R Crosstalk Via Binding to TSHR

Having established that TSHR and IGF1R could interact to increase signaling when activated by their respective ligands, it was important to determine whether activation of signaling by GO-Igs involved both receptors. However, it is difficult to demonstrate definitively that a single autoantibody is responsible for activating IGF1Rs using GO-Ig preparations since GO-Igs are polyclonal, that is, the population is comprised of multiple autoantibodies. Therefore, we [27] used the human monoclonal TSAb M22 for these initial studies. We found that the dose–response curve of M22 stimulation of HA secretion by GOFs was biphasic. Using linsitinib, an IGF1R kinase inhibitor, we showed that the high potency (low doses) phase of the curve was IGF1R-dependent, whereas the low potency (high doses) phase was independent of IGF1R. A model of the effects of M22 is illustrated in Figure 1. Although it is highly unlikely that a monoclonal antibody would bind to both TSHRs and IGF1Rs, We [8] used flow cytometry analysis with fluorescently tagged M22 and showed that M22 bound to TSHR-expressing cells but not to cells without TSHRs (but expressing IGF1Rs), and that TSHR binding was not inhibited by IGF1R blocking antibodies 1H7 and AF305 (Figure 2). Thus, M22 involves IGF1R in signaling even though it does not bind to IGF1R. This is a classic definition of one type of receptor crosstalk [29] that may involve direct receptor–receptor interaction or the presence of both receptors in the same signaling complex [37] (see below).

Although the data above show that M22 can activate TSHR/IGF1R crosstalk by binding only to TSHR, it was not clear whether other autoantibodies in GO-Ig preparations directly activated IGF1R. (A mouse monoclonal TSAb, KSAb1, also activated TSHR/IGF1R by binding only to TSHR—unpublished results.) We [11] used multiple approaches to determine whether GO-Igs directly activate IGF1R. First, we showed that a monoclonal anti-IGF1R blocking antibody, AF305, which completely inhibited IGF1 stimulation of HA secretion by GOFs, had no effect on GO-Ig stimulation of HA secretion (Figure 3). In contrast, another anti-IGF1R blocking antibody, 1H7, partially inhibited GO-Ig stimulation of GOFs [38] (Figure 3). These data could be explained by the receptor crosstalk paradigm. We hypothesize that 1H7 acts functionally differently than AF305 and inhibits TSHR/IGF1R crosstalk in addition to inhibiting binding to IGF1R. Second, we determined whether GO-Igs would stimulate IGF1R auto-phosphorylation, as auto-phosphorylation of IGF1R is the main mediator of IGF1R activation by IGF1 [39]. None of the GO-Ig preparations tested stimulated phospho-IGF1R production (Figure 4). Of note, the authors of two other studies measuring IGF1R phosphorylation as an indication of direct IGF1R activation by GO-Igs found only very small levels of phosphorylation, although one group concluded that there was no evidence of stimulating IGF1RAbs [12], whereas the other concluded that there may be evidence that a subset of GO patients express stimulating IGF1RAbs [13]. Similarly, Schwiebert et al. [40] found naturally occurring IGF1RAbs in young, overweight children, however, these IGF1RAb were antagonists not stimulators.

It is now appreciated that IGF1R can activate signaling cascades in the absence of demonstrable increases in phospho-IGF1R [39]. Therefore, it was important to use another readout of IGF1R activation to determine whether stimulating IGF1RAbs were present in GO-Ig preparations. Activation of AKT serine/threonine kinase 1 (AKT1) is a major step in signal transduction by IGF1R in many cells/tissues [39,41]. Marcus-Samuels et al. [42] found that 69% of GO-Igs caused minor activation of AKT in GOFs. Because stimulation of TSHR alone leads to AKT activation, Marcus-Samuels et al. further tested these GO-Igs in U2OS cells with no detectable endogenous TSHR expression. Although IGF1R knockdown inhibited robust IGF1-induced AKT activation by 65% [43], GO-Igs stimulated modest activation of AKT that was not affected by IGF1R knockdown.

Despite efforts to find IGF1R stimulation by GO-Igs in both recombinant cell lines and in human cells in primary cultures, we were not able to produce data that could be explained only by direct IGF1R stimulation. Thus, we are aware of no evidence of GO-Igs that directly bind to and activate IGF1R.

## 5. Concluding Remarks

We conclude that activation of TSHR/IGF1R crosstalk by GO-Igs is a critical mechanism involved in the pathogenesis of GO. A model of TSHR/IGF1R crosstalk is illustrated in Figure 5 where GO-Igs activate crosstalk by binding to and activating TSHRs on orbital fibroblasts. We found no evidence to support the hypothesis that some GO-Igs directly bind to and activate IGF1R. Therefore, we conclude that TSHR/IGF1R crosstalk is initiated by binding of GO-Igs to TSHR only.

TSHR is expressed in other extra-thyroidal tissues [44,45], and we hypothesize that TSHR/IGF1R crosstalk may occur in these tissues as well. In support of this hypothesis, we have observed TSHR’s association with IGF1R in primary cultures of human thyrocytes and in osteoblast-like cells that exogenously express TSHR [35]. Therefore, it is conceivable that TSHR/IGF1R crosstalk may be important in many tissues in which these receptors are natively expressed.

In contrast to stimulating IGF1RAbs, there is evidence that a small fraction of GO patients’ blood contains blocking IGF1RAbs [21]. These autoantibodies bind to but do not activate IGF1Rs. We admit that proving the negative hypothesis that there are no stimulating IGF1RAbs in the blood of GO patients is not possible. However, we think the accumulated data finding no direct evidence of stimulating IGF1RAbs in GO patients or in any human disease strongly suggests that stimulating IGF1RAbs do not exist in humans. Indeed, we think that if there were stimulating IGF1RAbs in humans, these antibodies would have marked effects on many tissues since IGF1Rs are expressed on many cells in the body. Moreover, we think it is the responsibility of the proponents of the hypothesis that stimulating IGF1RAbs are involved in GO pathogenesis to prove their hypothesis by isolating a monoclonal autoantibody from the blood of GO patients that binds to IGF1R, but not to TSHR, and activates IGF1Rs or TSHR/IGF1R crosstalk, or both.

We are aware of the exciting findings that a humanized monoclonal anti-IGF1R blocking antibody, teprotumumab, has been shown to be effective in the treatment of patients with GO [47]. If teprotumumab were found to behave like 1H7, then both of these antibodies could bind to IGF1Rs and inhibit TSHR/IGF1R crosstalk by a mechanism that has not yet been fully delineated. It is important to note that, as predicted, we have found that acute application of 1H7 will only partially inhibit GO-Ig stimulation of orbital cells in vitro because it only inhibits the IGF1R-dependent component, but not the IGF1R-independent component, of stimulation. Of course, it is possible that prolonged treatment with anti-IGF1R blocking antibodies in vivo may have additional actions that would completely inhibit stimulation.

As TSHR activation appears to initiate signaling by GO-Igs, it is likely that antagonism of TSHR would cause greater inhibition of stimulation by GO-Igs. Indeed, Place et al. [46] have shown this in in vitro studies using a small molecule, orally active drug-like antagonist of TSHR. Specifically, low doses of TSHR and IGF1R antagonists when added simultaneously caused complete inhibition of TSAb stimulation at lower than fully effective doses of either drug. Under certain conditions, it is likely antagonizing TSHR alone is sufficient to counter effects of TSAbs. Furthermore, a TSHR antagonist might retain efficacy at all concentrations of TSAbs, whereas an IGF1R antagonist may only inhibit at GO-Ig concentrations that activate crosstalk. Evans et al. also used the human monoclonal autoantibody with thyroid blocking activity K1-70 to inhibit TSHR signaling by several TSHR agonists in CHO cells transfected with TSHR and endogenously expressing IGF1R [48]. Furthermore, observations following expanded access administration of K1-70 to a single female patient with Graves’ disease, severe GO and locally advanced and distant metastatic well-differentiated follicular thyroid cancer demonstrated a dramatic improvement in the patient’s clinical activity score and exophthalmos [49,50]. K1-70 is currently in phase I clinical trials [51] and demonstrates the exciting potential of TSHR-targeting therapies.

Anti-TSHR antibodies [52], cyclic peptides [53] and small molecule ligands [54,55,56,57] are being developed as antagonists of TSHR activation with the potential to treat Graves’ hyperthyroidism and GO using the same drug. Furthermore, combination therapy with antagonists for TSHR and IGF1R might offer therapeutic benefits for patients due to dose reduction [46], and thereby minimize side effects of these drugs.

## Figures and Tables

**Figure 1 ijms-21-06561-f001:**
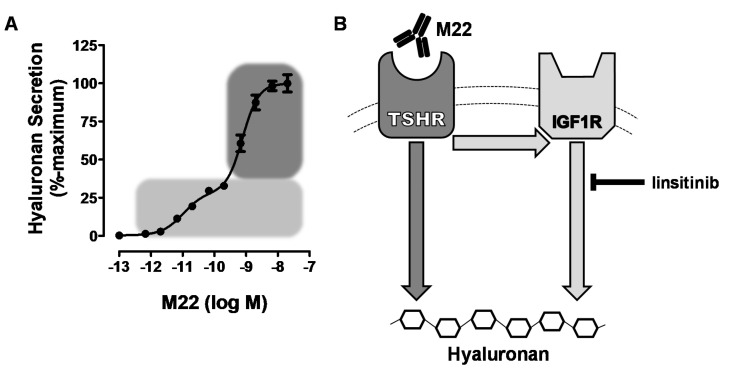
(**A**) model of thyroid-stimulating hormone receptor/insulin-like growth factor 1 receptor (TSHR/IGF1R) crosstalk initiated by M22 binding to TSHR on hyaluronan secretion by Graves’ orbital fibroblasts. Graves’ orbitopathy fibroblasts (GOFs) secrete hyaluronan following stimulation by M22, a human monoclonal TSHR-stimulating antibody. This response to increasing doses of M22 is biphasic (panel **A**), as shown by the light and dark gray regions of the curve. The light gray region is the IGF1R-dependent (high potency) phase, which was found by We to be inhibited by linsitinib, an IGF1R kinase inhibitor [27]. The dark gray region indicates the IGF1R-independent (low potency) phase of hyaluronan secretion, which occurs regardless of crosstalk with IGF1R. The cartoon in panel (**B**) depicts the two pathways that mediate TSHR-initiated hyaluronan secretion. The combination of these two pathways leads to full stimulation of hyaluronan. Adapted from [27].

**Figure 2 ijms-21-06561-f002:**
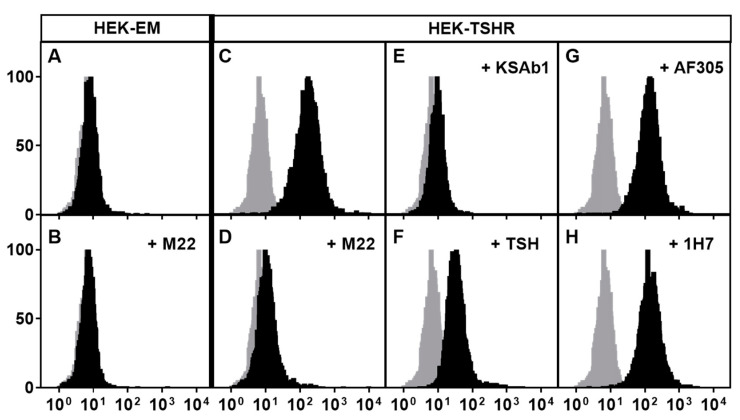
M22 specificity to TSHR demonstrated by competitive binding assay. The human monoclonal antibody M22 was coupled to the fluorescent probe Alexa-647 (Alexa-647-M22), and its binding to cell surface receptors was measured by flow cytometry. Binding of M22-647 was conducted in the presence of excess amounts of unlabeled TSHR agonists (M22, KSAb1, and TSH) and anti-IGF1R antibodies (AF305 and 1H7). Parental HEK293 (HEK-EM) cells do not express TSHRs but endogenously express IGF1Rs, and HEK-TSHR cells overexpress TSHRs and endogenously express IGF1Rs. Gray histograms (all panels) illustrate nonspecific cell autofluorescence. Black histograms (panels C–H) illustrate specific Alexa-647-M22 binding. Panels **A** and **B**: In HEK-EM cells, there is no specific binding. Panel **C**: specific binding that is totally inhibited (panel **D**) by an excess of unlabeled M22. Panel **E**: another unlabeled mouse monoclonal TSHR antibody, KSAb1, completely inhibits specific binding. Panel **F**: unlabeled TSH partially inhibits specific binding. Panels **G** and **H**: unlabeled anti-IGF1R blocking antibodies, AF305 and 1H7, have no effect on specific Alexa-647-M22 binding. Reprinted from [8].

**Figure 3 ijms-21-06561-f003:**
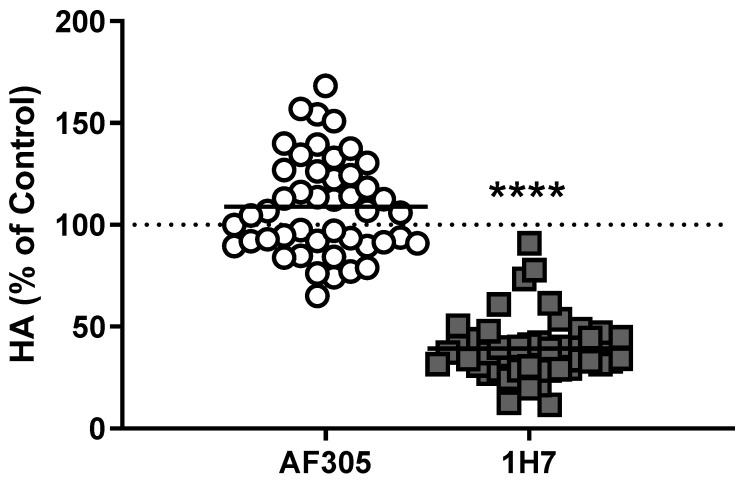
Anti-IGF1R blocking antibodies may or may not inhibit GO-Ig stimulation of hyaluronan secretion by Graves’ orbital fibroblasts. GOFs were stimulated with purified GO-Igs in the presence of IGF1R blocking antibodies, AF305 (white circles) and 1H7 (gray squares). Data were normalized to their own control (dotted line). Each data point represents GO-Igs from one patient, and black bars indicate the averaged means of all stimulatory GO-Igs. Both AF305 and 1H7 completely inhibit binding of IGF1 to IGF1Rs [11]. While AF305 had no effect on GO-Ig stimulation of hyaluronan (HA) secretion, 1H7 partially inhibited GO-Ig stimulation of HA secretion. These findings are consistent with the idea that AF305 does not affect TSHR/IGF1R crosstalk, whereas 1H7 inhibits crosstalk. Reprinted from [11]. **** *P* < 0.001 vs. GO-Ig control by Student’s t-test.

**Figure 4 ijms-21-06561-f004:**
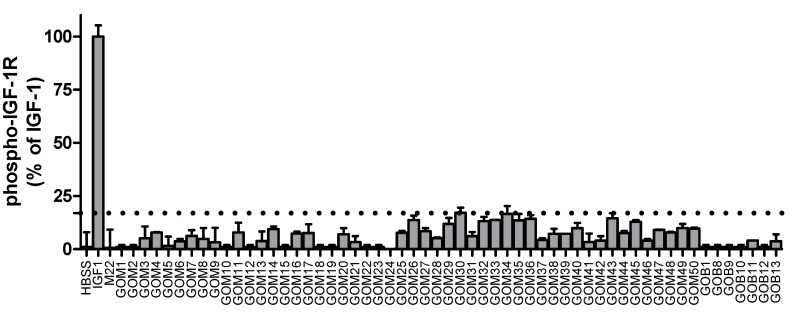
IGF1Rs are not activated by GO-Igs in Graves’ orbital fibroblasts. IGF1, the human monoclonal antibody M22 and 57 GO-Ig preparations were used to determine whether GO-Igs stimulated the phosphorylation of IGF1R in GOFs. Rapid autophosphorylation of IGF1R is the main initiator of IGF1R signaling. As expected, IGF1 stimulated a robust increase in phospho-IGF1R, but M22 had no effect. None of the 57 GO-Ig preparations tested increased phospho-IGF1R. Reprinted from [11].

**Figure 5 ijms-21-06561-f005:**
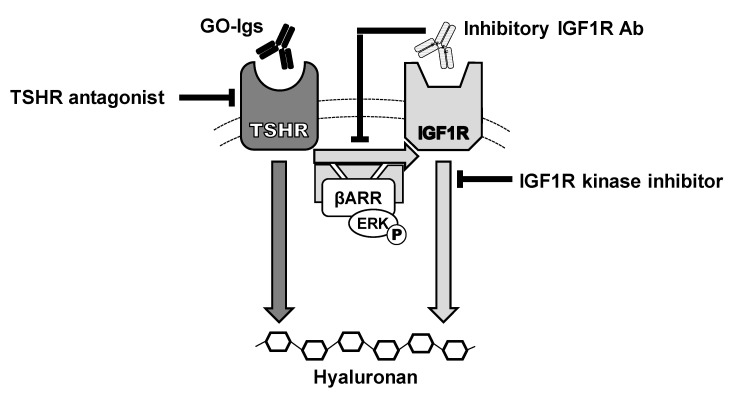
Model of the effects of IGF1R and TSHR antagonists on GO- immunoglobulin (Ig) stimulation of hyaluronan secretion by Graves’ orbital fibroblasts. GO-Igs bind directly to and activate TSHR, stimulating two signaling pathways: IGF1R-independent (dark gray arrow) and IGF1R-dependent (light gray arrows), leading to secretion of hyaluronan. TSHR/IGF1R crosstalk is initiated by binding of GO-Igs to TSHR and requires TSHR and IGF1R to be in close proximity within a signalosome. Synergistic interactions between TSHR and IGF1R occur rapidly at phosphorylation of ERK, demonstrating that crosstalk between TSHR and IGF1R occurs early in the signaling cascade, proximal to the receptors [30]. β-arrestin 1 (βARR) is essential for GO-Ig stimulation of ERK phosphorylation, and hyaluronan secretion and proximity ligation assays in GOFs demonstrated that β-arrestin 1 physically scaffolds TSHRs and IGF1Rs in a protein complex [35]. These pathways appear not to involve cAMP, which has been considered a canonical pathway of TSHR in thyrocytes. However, recent evidence has demonstrated β-arrestin-mediated signaling in thyrocytes and osteoblast-like cells also. Some inhibitory IGF1R antibodies (IGF1RAb), such as 1H7 and probably teprotumumab, bind to IGF1Rs and inhibit activation of IGF1R by activated TSHR via crosstalk. TSHR antagonists, including drug-like small molecule inhibitors, anti-TSHR antibodies and cyclic peptides via immune hyposensitization inhibit signaling initiated by GO-Ig activation of TSHR, and thereby totally abolish both pathways of stimulation of hyaluronan secretion. Combination therapy with TSHR and IGF1R antagonists could minimize drug side effects and, therefore, may have therapeutic benefits [46].

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
