# Peer review of "Is There Evidence for IGF1R-Stimulating Abs in Graves’ Orbitopathy Pathogenesis?"

_ijms, 2020, doi:10.3390/ijms21186561_

Round 1

Reviewer 1 Report

This is an interesting paper that is very well-written.  I have no major concerns.

Minor concerns:

lines 18-19:  Add the keywords.

line 98:  Change to "et al.".

lines 304-305:  I am not sure that this degree of conjecture is appropriate in a scientific publication such as this.

Author Response

Minor concerns:

Point 1: lines 18-19:  Add the keywords.

Response 1: We added keywords (lines 19-20)

Point 2: line 98:  Change to "et al."

Response 2: We changed to et al. (line 99)

Point 3: lines 304-305:  I am not sure that this degree of conjecture is appropriate in a scientific publication such as this.

Response 3: We changed the text as follows, “ If teprotumumab were found to behave like 1H7, then both of these antibodies could bind to IGF1Rs and inhibit TSHR/IGF1R crosstalk by a mechanism that has not yet been fully delineated.”  (lines 308 – 310)

Because we are proposing an alternative hypothesis, it is important to explain how this alternate view aligns with existing clinical data. We wish to keep the above for that reason.

Also note we have corrected the following two typos:

line 100: changed "IGF1RAb" to "IGF1RAbs"

line 342: In the abbreviations section, changed "Stimulatory IGF1R Antibodies" to "IGF1R Antibodies"

Reviewer 2 Report

The present article reviews the evidence of direct stimulation of IGF1 receptors (IGF1Rs) by autoantibodies in Graves’ orbitopathy (GO) pathogenesis. The manuscript has come from an established group of researchers who have contributed significantly in the concerned field of research. The major outcome of the review article was to provide evidence-based information that shows GO pathogenesis does not involve directly stimulating IGF1RAbs.

Major:

  1. Overall, this is a well-written review article and the authors approach to sum up all those relevant literatures to put together to critically discuss the mechanistic approach to understand the role of IGF1Rs in relation to GO pathogenesis. Some inconsistencies regarding presentation /citation of the authors own work have been noted. Christine C. Krieger, the first author of the present manuscript cited many of her own references using third person plural number (Krieger et al) and simultaneously the detailed discussion of the cited references have been written using first person plural number (We). Page 3, lines 138-141 and elsewhere. Choose any of the styles and make consistent use of it.

  1. Figures (1-4) presented in the manuscript have been either reprinted or adapted with minor changes incorporated from previously published articles by the same group of authors. The authors need to check regarding obtaining any copyright-related permission that might be required for such reproductions.

  1. The derivation of the hypothetical model regarding the units of critical puzzle as to how IGF1R Abs are linked to GO pathogenesis functionally relied on the data obtained from the novel assay quantifying endogenous IGF1R-Abs in humans and testing their biological effects in vitro. Indeed the authors have acknowledged this limitation (Page 9, lines 309-311). This limitation alone weakens the validity of the model in a clinical setting and the authors may consider rephrasing the title of the manuscript orienting the readers to make them quiet aligned with the present status of the in vitro studies.

Minor:

  1. ‘et al.’ used throughout the manuscript needs to be italicized.
  2. Page 2, line 93:’ IGF1RAbs’ has been wrongly written as ‘IFG1RAbs’.
  3. Page 3, line 114: By mentioning ‘IGF1Abs’, do you mean IGF1RAbs?

Author Response

Major:

Point 1: Overall, this is a well-written review article and the authors approach to sum up all those relevant literatures to put together to critically discuss the mechanistic approach to understand the role of IGF1Rs in relation to GO pathogenesis. Some inconsistencies regarding presentation /citation of the authors own work have been noted. Christine C. Krieger, the first author of the present manuscript cited many of her own references using third person plural number (Krieger et al) and simultaneously the detailed discussion of the cited references have been written using first person plural number (We). Page 3, lines 138-141 and elsewhere. Choose any of the styles and make consistent use of it.

Response 1: We changed “Krieger et al.” to “We.” (lines 133, 140, 147, 175, 181, 192, and 215)

Point 2: Figures (1-4) presented in the manuscript have been either reprinted or adapted with minor changes incorporated from previously published articles by the same group of authors. The authors need to check regarding obtaining any copyright-related permission that might be required for such reproductions.

Response 2: There are no copyright issues because we hold the copyrights to all our publications as a federal government lab per agreement between the journals and the US federal government.  

Point 3: The derivation of the hypothetical model regarding the units of critical puzzle as to how IGF1R Abs are linked to GO pathogenesis functionally relied on the data obtained from the novel assay quantifying endogenous IGF1R-Abs in humans and testing their biological effects in vitro. Indeed the authors have acknowledged this limitation (Page 9, lines 309-311). This limitation alone weakens the validity of the model in a clinical setting and the authors may consider rephrasing the title of the manuscript orienting the readers to make them quiet aligned with the present status of the in vitro studies.

Reponse 3: In addition to the IGF1R antibody assay, we present examples of negative data using sera derived from GO patients [42]. These sera were used in experiments very similar to clinical assays used to make diagnoses. We also tested our model using GO sera on cells derived from GO patients. We argue that the use of GO patient sera and cells support the clinical applicability of our model. Our other objective is to point out the dearth of data supporting the model behind the teprotumumab trials.

To highlight this, we changed the title to “Is there evidence for IGF1R-stimulating Abs in Graves’ orbitopathy pathogenesis” (lines 3-4)

Minor:

Point 1: ‘et al.’ used throughout the manuscript needs to be italicized.

Response 1: We italicized all use of et al. (lines 138, 158, 228, 251, 253, 314, and 320)

Point 2: Page 2, line 93:’ IGF1RAbs’ has been wrongly written as ‘IFG1RAbs’.

Response 2: We changed to “IGF1RAbs” (line 94) and thank the reviewer for catching this typo.

Point 3: Page 3, line 114: By mentioning ‘IGF1Abs’, do you mean IGF1RAbs?

Response 3: That is correct, and we changed to “IGF1RAbs” in the text (line 115).

Also note we have corrected the following two typos:

line 100: changed "IGF1RAb" to "IGF1RAbs"

line 342: In the abbreviations section, changed "Stimulatory IGF1R Antibodies" to "IGF1R Antibodies"